REGISTERED REPORT PROTOCOL

# Perceptions on respectful maternity care in Sri Lanka: Study protocol for a mixed-methods study of patients and providers

**Malitha Patabendige**[1‡¤*], **Suneth Buddhika Agampodi**[2], **Asanka Jayawardane**[3], **Denagamage Jayamini Wickramasooriya**[4], **Thilini Chanchala Agampodi**[2]

**1** Obstetrics and Gynaecology Wards- 07 and 08, Castle Street Hospital for Women, Colombo, Sri Lanka, **2** Faculty of Medicine and Allied Sciences, Department of Community Medicine, Rajarata University of Sri Lanka, Mihintale, Sri Lanka, **3** Faculty of Medicine, Department of Obstetrics and Gynaecology, University of Colombo, Colombo, Sri Lanka, **4** Teaching Hospital, Karapitiya, Galle, Sri Lanka

‡ This author contributed mainly to this work.
¤ Current address: Base Hospital, Pottuvil, Sri Lanka
* mpatabendige@gmail.com

## Abstract

### Background

Over the past few decades, interest in providing and measuring Respectful Maternity Care (RMC) has increased markedly. Sri Lanka is reportedly shown to have better maternal health statistics and studies on quality improvement are lacking in this unique population. We aim to describe healthcare providers' perceptions and women's expectations, perceptions and their gaps in service provision regarding RMC in Sri Lanka.

### Methods

A descriptive cross-sectional study with a mixed-methods approach comprising of qualitative component followed by a quantitative component will be conducted in Castle Street Hospital for Women (CSHW) and De Soysa Hospital for Women (DSHW), Colombo, Sri Lanka. Healthcare providers (HCP- doctors, nurses and midwives) and vaginally delivered postnatal women (in postnatal wards and postnatal well-baby clinics) will be recruited through convenience sampling. In-depth interviews will be conducted with each of the four categories. Thematic analysis will be adopted to analyze qualitative data and the findings will further be used to improve the quantitative phase questionnaires. Self-administered questionnaire will be administered to a 378 vaginally delivered postnatal women [quota sampling across ten wards], exploring demographic details, and maternal opinion on various aspects of RMC. Locally validated Women's Perceptions of RMC tool (WP-RMC) will also be used to measure the level of RMC among these postnatal women along with the questionnaire 01. Qualitative findings will be used for cognitive validation of the WP-RMC into the Sri Lankan setting.

**Data Availability Statement:** All relevant data from this study will be made available upon study completion.

**Funding:** The author(s) received no specific funding for this work.

**Competing interests:** The authors have declared that no competing interests exist.

## Discussion

This study will explore HCP's and women's expectations, perceptions and their gaps in service provision regarding RMC in two maternity hospitals in Sri Lanka. Assessment of the quality of care with regards to RMC have not been reported previously in this setting.

## Introduction

Improving quality of maternal and newborn care in healthcare facilities is identified as a key factor in further reduction of maternal deaths [1]. Respectful maternity care (RMC) plays a main role considering quality of services. The World Health Organization defines RMC as "care organized for and provided to all women in a manner that maintains their dignity, privacy, and confidentiality, while ensuring freedom from harm and mistreatment, and enables informed choice and continuous support during labor and childbirth" [2,3].

At present, "disrespectful maternity care" is common and widespread worldwide [4,5]. It may include physical or sexual violence such as hitting, undue restraints, and unnecessary exposure of the woman's genitalia and rough vaginal examinations. Women at childbirth often become victims of verbal abuse and may experience "neglectful care" consisting of restricted mobility, leaving laboring alone, unsupportive birth attendants, and withholding of food and/or drink [5–7]. Women might experience serious injuries and mistreatment both physically and emotionally. Higher rates of maternal as well as neonatal and infant mortality and morbidity is associated with disrespectful maternity care in LMICs [4]. Existence of RMC has good short-term and long-term outcomes on the mother and her family [8]. A review by Shakibazadeh et al., in 2018 have concluded that the concept of RMC is broader than just reduction of disrespectful care or minimising mistreatment of women during childbirth [9]. Twelve domains of RMC have derived and described from the qualitative findings of this review and presented in Table 1.

Sri Lanka is a role model for maternity care among LMIC countries with exemplary achievements [10–12]. The maternal mortality ratio (MMR) for Sri Lanka is 32.0 per 100,000 live births in 2018 and it is an example of how low-cost, context-specific interventions could lead to exponential improvements in health indices. Nonetheless, maternal suicides play a

**Table 1. Twelve domains of the respectful maternity care.**

| Domains of the respectful maternity care |
| --- |
| 1) Being free from harm and mistreatment |
| 2) Maintaining privacy and confidentiality |
| 3) Preserving women's dignity |
| 4) Prospective provision of information and seeking informed consent |
| 5) Ensuring continuous access to family and community support |
| 6) Enhancing quality of physical environment and resources |
| 7) Providing equitable maternity care |
| 8) Engaging with effective communication |
| 9) Respecting women's choices that strengthens their capabilities to give birth |
| 10) Availability of competent and motivated human resources |
| 11) Provision of efficient and effective care |
| 12) Continuity of care |

Source: Shakibazadeh E, et al. Respectful care during childbirth in health facilities globally: A qualitative evidence synthesis. BJOG An Int J Obstet Gynaecol. 2018;125(8):932–42.

significant role in maternal mortality in the country [13] reflecting poor psychosocial wellbeing of pregnant and postpartum women. Being in the fourth phase of obstetric transition [14] improvement of quality of care is of high priority in the country in order to further reduce maternal deaths. The importance of RMC for the Sri Lankan context is further reflected by the high cesarean delivery rates [40.5%] in the country and studies and assessments on the quality of care and quality improvement in Sri Lankan context is scarce. Furthermore, a survey among obstetricians have shown that a poor attitude 58.8% of Sri Lankan obstetricians were not allowing the labour companion and this can be considered as a very unfortunate situation in a setting like Sri Lanka [15]. Therefore, Sri Lankan setting is unique and Colombo contains the country's two largest maternity hospitals (study settings) and are considered as model institutions in providing quality maternal healthcare [16]. Therefore, considering the above facts, this study aims to cover a timely need in Sri Lanka despite numerous research on this regard in neighboring India and other LMIC [17–20].

This study aims to explore and describe the expectation- perception gap in RMC during labour and childbirth among women. The planned study is a novel concept for the Sri Lankan obstetric setting and is expected to yield more insight on recommendations to promote RMC.

## Methods

### Study design

A descriptive cross-sectional study with a mixed-methods approach comprising of qualitative component [in-depth interviews] followed by quantitative component [two self-administered questionnaires to vaginally delivered postnatal women].

### Study setting

Study setting will be CSHW and DSHW, Colombo, Sri Lanka that are two major tertiary care maternity hospitals in Sri Lanka. Approximate annual delivery rate in CSHW is 11000 live-births and that of DSHW is 8000. DSHW and CSHW are the first maternity hospitals situated in Sri Lanka and that they are considered as model institutions [16]. Therefore, any deficiency in the quality of care in these two referral hospitals situated in the capital Colombo can possibly represent much wider deficiencies elsewhere of the country. The RMC and mistreatment during labour have been extensively studied in other LMIC like India [17,18,21,22]. However, as a country with a relatively better maternal and neonatal statistics and being rewarded as a role model to other LMIC [11,12], this Sri Lankan study setting is different compared to other studies. Studies aiming strategies for quality improvement is a timely need in Sri Lanka. The ethnic breakdown of patients in these hospitals which would be valuable as it has a good representation of minority groups. In CSHW, there are five labour wards, four operating tables, one six-bed intensive care unit and three well-baby clinics per week. While in DSHW, there are five labour wards, four operating tables, one high-dependency unit, one three-bed intensive care unit and five well-baby clinics per week. Health staff involved in care provision includes approximately 200 medical officers, 300 nurses, 70 midwives, and 400 support staff in both hospitals.

### Study population

HCP working in both hospitals, vaginally delivered women staying in postnatal wards and vaginally delivered women coming to well-baby clinics in CSHW and DSHW, Colombo, Sri Lanka, during the study period.

## Qualitative phase

Study will be started with the qualitative phase in CSHW and DSHW, Colombo, Sri Lanka. In-depth interviews will be conducted covering main areas of RMC to explore women's expectations and perceptions of RMC. As well as, RMC as perceived by the HCP.

## Study participants for the qualitative phase

For the qualitative phase will belong to the following categories;

1. Postnatal mothers in the ward and clinic: Mothers who have had their birth experience and staying in postnatal wards and mothers coming to well-baby clinics, will be invited.

2. HCP working in antenatal wards, labour wards, and postnatal wards in CSHW and DSHW will be invited.

## Inclusion and exclusion criteria for the qualitative phase

**Criteria for women.** *Inclusion criteria.*

a. Age at least 18 years old or more at the time of delivery.

b. All the vaginally delivered postnatal women with an uncomplicated antenatal period.

c. Not known to have fetal anomalies.

**Criteria for HCP.** *Inclusion criteria.* Three categories of HCP: doctors, nurses and midwives working in CSHW and DSHW during the study period.

## Sampling method and sample size in the qualitative phase

Convenience sampling will be used to obtain an adequate representation of different types of HCP categories and mothers representing different ethnic groups, parities, educational and socio-economic status. The number of participants will be increased until data saturation occurs in each of the four arms of study participants- HCP [doctors, nurses and midwives] and postnatal mothers. As there are only four postnatal wards [4/10] belonging to units that are practicing the labour companionship, 50% of HCP and postnatal women will be recruited from these four wards and the rest from the other six postnatal wards.

## Participant recruitment, interview guides and data collection procedure in the qualitative phase

Participants will be recruited maintaining an obstetric clinical diversity with regards to the mode of delivery and the neonatal outcome to yield representativeness during sample selection. If the postnatal women are giving insufficient information while staying the hospital possibly due to social response bias, postnatal interviews will be extended into the postnatal well-baby clinics where mothers will feel less threatened to give less socially desirable responses as they are not bound to the labour ward anymore. Recruiting and interviewing will be continued until the phenomena under the study are well understood and data saturation is reached.

Interviewer guides will be prepared according to the standard methods [23] separately for postnatal mothers [S1 File] and HCP [S2 File]. Trained accredited translators will translate all interview guides from the English language to Sinhala and Tamil languages. All dimensions of RMC will be considered in the preparation of the interview guides. With regards to the HCP,

their perceived knowledge, practice and attitudes to promote RMC will be qualitatively assessed. Results of this part can improve the WP-RMC questionnaire validation and has the potential to generate providers' perceptions on RMC. These interviewer guides were qualitatively pre-tested with a group of HCP (n = 10) and postnatal women (n = 10) in another hospital.

All interviews will be conducted in the first language of the participant. A female doctor and/or two female nurses having trained on the qualitative methods will conduct in-depth interviews. Whenever necessary, a separate data collecting female medical officer will conduct Tamil language interviews. Interviews will be carried out in a quiet place in the ward without disturbing the routine ward activities. The interviews will be conducted according to the Family Health International (FHI) guidelines [24] and other relevant texts available in the literature [25]. Whyte *et al*, in 1987 developed a six-point directiveness scale to help researches analyze their interviewing technique. The PI will go through the Whyte's defectiveness scale [26] before conducting each interview and after the interview will analyze the interviewing technique using these criteria. As some informants are verbose than others, to maintain control of the interview, investigators will follow Patton's strategies [27], the three strategies developed by Patton [1987] for maintaining control of the interview.

All interviews will be audio-recorded with consent to avoid transcription and translation errors. Field notes will be made during and immediately after the interviews. Each in-depth interview will last approximately 30–60 minutes. Data will be expanded immediately and transcribed within two weeks. Transcripts will be validated for accuracy by the local research team at CSHW and DSHW.

## Methods to ensure trustworthiness and quality control of the qualitative components of the study

1. The following steps will be taken to ensure the quality and trustworthiness of this study component. All data collection tools will be designed according to FHI guidelines [24].

2. Triangulation minimizes bias due to chance associations and systematic biases due to a specific method in qualitative studies [28]. This will be achieved by comparison of data obtained through several in-depth interviews and different sources (postnatal mothers and several levels of HCP).

3. Respondent validation which is considered as the strongest available check on the credibility of qualitative research [29] will be carried out as a participant checking in interviews.

4. The quality of data collection will be maintained by using checklists in all field visits. Interviewers will be trained and supervised.

5. To enhance reflexivity [29], a research diary will be maintained during data collection and analysis to document the assumptions, biases, and reactions to events occurring during the period of research and will discuss these issues in the presentation of the study. This will enable to express reflexivity of the study.

6. Data collection and analysis methods will be done paying attention to rigor and transparency of the procedures.

7. Investigators trained in qualitative analysis and knowledgeable on the concept of labour and childbirth will participate in data analysis. They will collectively perform the thematic analysis following independent coding and consensus and all authors will have consensus

simultaneously to improve the credibility of the themes. When different opinions arise, these will be discussed among the three investigators and a consensus will be generated.

## Quantitative phase

This will be the subsequent phase and data collection instruments will be modified according to the results of the qualitative phase. This phase will also be focused to describe the expectation- perception gap in respectful maternity care during labour and childbirth among women undergoing childbirth in CSHW and DSHW, Colombo, Sri Lanka.

Qualitative findings about RMC will be used for cognitive validation of the Women's Perceptions of RMC (WP-RMC) local versions and the items in WP-RMC will be properly translated in a context-specific manner with aid of these qualitative inputs.

## Inclusion and exclusion criteria for postnatal women in the quantitative phase

**Inclusion criteria.**

a.  Age at least 18 years old or more at the time of delivery.

b.  All the vaginally delivered postnatal women with an uncomplicated antenatal period.

c.  Not known to have fetal anomalies.

## Sample size calculation for the quantitative phase

The sample size will be calculated to assess the prevalence of disrespectful care using the formula for estimation of the proportion of a binary outcome;

$$n = \frac{Z^2 \frac{\alpha}{2} P(1-P)}{d^2}$$

d -Margin of error = 0.05,
Z&/2–95% confidence interval,
P- Proportion (P) for disrespectful care as 0.7131. The overall pooled prevalence of disrespectful maternity care that was found in an Indian systematic review and meta-analysis was 71.31% [30]. The calculated sample size is 315 pregnant mothers. With an estimated 20% non-response rate, the final minimum sample size would be 378 participants. A higher non-response rate was chosen considering the sensitive nature of the problem that will be investigated in this vulnerable population.

**Sampling method for the quantitative phase.**   There are five postnatal wards in CSHW and five in DSHW. Quota sampling will be conducted and approximately 40 vaginally delivered postnatal women with an uncomplicated antenatal period will be recruited from each of the 10 postnatal wards to yield the minimum sample size.

**Participant recruitment, questionnaires and data collection procedure for the quantitative component.**   All the women will be given self-administered questionairre-01 [S3 File] and self-administered WP-RMC. Questionnaire 01 was prepared after reviewing the relevant literature and childbirth experience questionnaire [31–33]and consists of 15 questions in a five-point Likert scale ['Strongly agree' to 'Strongly disagree'] focusing on maternal opinion on various aspects of RMC. There were three questions about her sense of control, perception of

pain and sense of security on a 1–10 visual analogue scale. This will be further modified using the results of the qualitative study. Face validation was done for the questionnaire with the participation of obstetricians, public health specialists. Pre-testing was conducted among a sample (n = 25) of postnatal women in another hospital.

The Iranian origin WP-RMC questionnaire is a recently developed, valid and a reliable tool to assess women's perception regarding RMC amongst vaginally delivered women [34]. The tool is self-administered and contain 19 items. Among other tools available to assess women's perception on RMC [35–37] and patient centered care [20], the WP-RMC tool seems to be more robust since it's development and validation has been conducted according to WHO RMC recommendations and a thorough review on the available literature on RMC [34]. Afulani et al have recently developed 27-item tool, person-centered maternity care scale to assess RMC in India and this is also another robust tool developed for this purpose [17,18]. However, WP-RMC is a relatively short, easy to understand hence can be administered in large scale. Therefore, investigators preference was to choose WP-RMC in the present study.

WP-RMC needs to be translated and validated into Sinhala and Tamil languages in the Sri Lankan context. For this purpose, the previously mentioned steps of translation and validation will be performed [38,39]. Initially the original English questionnaire will be translated into Sinhala and Tamil by a Sinhala and Tamil native accredited translators separately [forward translation]. This version will be reviewed by two separate Sinhala [native language] and Tamil [native language] obstetricians. The available items of the original tool will be modified, removed or new items may be added to culturally adapt the tool using the results of the initial qualitative component. This integration of qualitative findings is aimed to strengthen its cognitive validity.

Furthermore, WP-RMC Sinhala and Tamil versions will be given to 20 experts [obstetricians and public health specialists] and content validity index will be calculated. After necessary modifications by the experts, the final translations of Sinhala WP-RMC (WP-RMC-S) and Tamil WP-RMC (WP-RMC-T) will be produced. Internal consistency and test re-test reliability (administering the tool for 20 mothers from each ethnic group two weeks apart) of the tool will be assessed for both translations. Considering the 5–10 times the number of observed variables into the instrument, two samples of 190 women delivered vaginally at term will be recruited from Sinhala and Tamil speaking ethnicities separately for assessment of structural validity. The study settings will be selected according to the ethnic distribution of maternity ward admissions for each translations. Depending on the components of RMC, we will perform exploratory and/or confirmatory factor analysis to assess structural validity. Criterion validity of the WP-RMC-S and WP-RMC-T tools will be assessed with Edinburgh postnatal depression scale (EPDS). For structural validity of WP-RMC tools if data reduction is needed, exploratory factor analysis with maximum likelihood method using direct oblimin rotation will be conducted using SPSS.

## Data analysis

The PI will initially familiarize himself by reading and re-reading the notes and transcripts. Three investigators will participate in data analysis. Two investigators will independently code data and a coding scheme will be developed and used to code all interview transcripts. The process will be conducted manually as data will be analyzed in the native language to minimize data loss.

As the dimensions of RMC are already identified, data analysis will include a framework approach [40]. However, initially thematic analysis will be conducted which will allow the authors to develop a rich thematic description regarding the issues around childbirth and

labour experience. It will allow investigators to identify, analyze and report common themes with a minimum level of interpretation from the investigator [41]. The identified subthemes will be then fitted into the available frame work of RMC. As RMC is a new study area for the Sri Lankan context, the PI will be sensitive to and vigilant of new themes that could emerge apart from the available framework. The new themes will be added on consensus of the investigators.

The analytic process will include the following six steps given by Braun et al [42]. All steps need the consensus of all investigators.

1. Preparing the transcript;

2. Familiarizing data;

3. Generating initial codes by marking words or sentences relevant to the topic of inquiry;

4. From the codes, draft themes will be generated in an iterative process; independent coding will be performed by two investigators;

5. The main categories and sub-categories of the themes will be identified;

6. The results will be written up.

Statistical analysis of the validation part including assessment of the psychometric properties has been mentioned above. In the quantitative component, descriptive statistics and basic demographic characteristics of the quantitative data will be analyzed to see any deviations. Continuous variables with normal distributions will be presented as means with standard deviations. Discrete numeric variables and ordinal variables were presented as medians with interquartile ranges. Nominal and categorical variables will be presented as proportions. The expectation-perception gap will be described using relevant data analysis and presentation techniques. Significance will be declared at p-value $<0.05$. Standard statistical methods will be followed.

### Duration of the study

The duration of the study will be approximately 12 months.

### Data maintenance storage and disposal

Data will be entered into a data collection sheet and confidentially stored in an ongoing computer database.

### Plan of presentation of results

Findings will be presented at academic symposia and published in indexed peer-reviewed journals.

### Ethical considerations

Ethical approval was obtained from the Ethical Review Committee [ERC/256/05/2019], CSHW, Colombo, Sri Lanka.

### Work plan and timelines

Currently ethics and administrative approvals have been obtained. Data collection is yet to be started soon and will take another four months. Data analysis and writing of the paper will take another three months approximately.

## Strengths and limitations of the study

This research will add new knowledge to the existing literature filling an important research gap of RMC in LMIC settings. It will also lead to an increased understanding of a hidden area that has previously gained limited attention- the structural, cultural and social context behind the link between childbirth, mothers' expectations and staff interactions in our setting. Collecting data at several levels and through triangulation between data sources, informant groups, and theories ensures comprehensiveness and quality of the data gathered. The mixed method approach will ensure rigor and quality of evidence generated from this study.

Considering the limited studies available, the poor attitudes [15] and high cesarean section rates in the country, this study will shed light on policy implementations with respect to quality maternal care in Sri Lanka. The findings would be beneficial for other LMICs with similar contexts and the culturally adapted tools could be used in similar settings to measure RMC. The study population will be limited to Colombo which is not generalizable into entire Sri Lanka making issues with its external validity.

## Supporting information

**S1 File. Interviewer guide for the postnatal mothers.**
(PDF)

**S2 File. Interviewer guide for the healthcare providers.**
(PDF)

**S3 File. English version of questionnaire-1 to assess socio demographic factors and expectations of RMC.**
(PDF)

**S4 File. Standards for Reporting Qualitative Research (SRQR) checklist [43].**
(DOCX)

**S5 File. Labour study ERC letter.**
(PDF)

## Author Contributions

**Conceptualization:** Malitha Patabendige, Suneth Buddhika Agampodi, Thilini Chanchala Agampodi.

**Data curation:** Malitha Patabendige.

**Formal analysis:** Malitha Patabendige.

**Investigation:** Malitha Patabendige.

**Methodology:** Malitha Patabendige, Suneth Buddhika Agampodi, Asanka Jayawardane, Thilini Chanchala Agampodi.

**Project administration:** Malitha Patabendige, Asanka Jayawardane.

**Resources:** Malitha Patabendige, Asanka Jayawardane, Denagamage Jayamini Wickramasooriya.

**Software:** Malitha Patabendige.

**Supervision:** Malitha Patabendige, Suneth Buddhika Agampodi, Thilini Chanchala Agampodi.

**Validation:** Malitha Patabendige, Asanka Jayawardane.

**Visualization:** Malitha Patabendige.

**Writing – original draft:** Malitha Patabendige.

**Writing – review & editing:** Malitha Patabendige, Suneth Buddhika Agampodi, Asanka Jayawardane, Denagamage Jayamini Wickramasooriya, Thilini Chanchala Agampodi.

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
