## [Decision Letter · Decision Letter 0]

25 Aug 2020

PONE-D-20-03776

Maternal and Healthcare Providers’ Perceptions of Quality of Care during Labour and Childbirth: Study Protocol for a Mixed-Methods Study.

PLOS ONE

Dear Dr. Patabendige,

Thank you for submitting your manuscript to PLOS ONE. After careful consideration, we feel that it has merit but does not fully meet PLOS ONE’s publication criteria as it currently stands. Therefore, we invite you to submit a revised version of the manuscript that addresses the points raised during the review process.

We look forward to receiving your revised manuscript.

Kind regards,

Marzia Lazzerini, PhD

Academic Editor

PLOS ONE

Journal Requirements:

2. Please include a copy of the interview guide used in the study, in both the original language and English, as Supporting Information, or include a citation if it has been published previously.

3. Please include additional information regarding the survey or questionnaire used in the study and ensure that you have provided sufficient details that others could replicate the analyses. For instance, if you developed a questionnaire as part of this study and it is not under a copyright more restrictive than CC-BY, please include a copy, in both the original language and English, as Supporting Information.  If the original language is written in non-Latin characters, for example Amharic, Chinese, or Korean, please use a file format that ensures these characters are visible.

4. Please state whether you validated the questionnaire prior to testing on study participants. Please provide details regarding the validation group within the methods section.

6. Please include a caption for figure 1.

Additional Editor Comments (if provided):

Please read carefully the feedbacks from the referees and provide a point by point answer.

Please use the appropriate reporting guidelines checklists, such as STROBE and COREQ or SRQR (for qualitative research)

Please also specify if the study has been registered in any platform

Please also specify study timelines

Reviewers' comments:

Reviewer's Responses to Questions

**Comments to the Author**

1. Does the manuscript provide a valid rationale for the proposed study, with clearly identified and justified research questions?

Reviewer #1: Partly

Reviewer #2: Yes

2. Is the protocol technically sound and planned in a manner that will lead to a meaningful outcome and allow testing the stated hypotheses?

Reviewer #1: Partly

Reviewer #2: Partly

3. Is the methodology feasible and described in sufficient detail to allow the work to be replicable?

Reviewer #1: No

Reviewer #2: Yes

4. Have the authors described where all data underlying the findings will be made available when the study is complete?

Reviewer #1: No

Reviewer #2: Yes

5. Is the manuscript presented in an intelligible fashion and written in standard English?

Reviewer #1: No

Reviewer #2: Yes

6. Review Comments to the Author

You may also provide optional suggestions and comments to authors that they might find helpful in planning their study.

Reviewer #1: The study protocol has the objective of exploring health professionals and women’s perceptions regarding the quality of care during labour and childbirth in Sri Lanka. This is an interesting project that would make an important contribution to the literature. I appreciated the opportunity to read the manuscript and specific comments are stated below.

First, the research questions are not clearly stated on last paragraphs of background section, but on lines 102-104 and 113-115; also, research questions are not in line with the following section: Objectives of the study (line 177). Please revise. I would suggest the use of reporting guidelines checklists, such as STROBE and COREQ or SRQR (for qualitative research). It would help to improve the hole paper comprehension. In addition, reference numbering is confusing and not organized in the order that articles were cited in the text (for example, in the first paragraph [lines 78-85] were cited references 8, 11 and 21). I recommend that the authors closely review and edit the paper.

Second, on methods would be helpful to provide more information regarding setting and specific hospital units described (example, well-baby clinic). I am assuming that De Soysa Hospital for Women (DSHW) should be also mentioned on lines 467 and 468, please review and edit if necessary.

Third, please clarify recruitment procedures. For example, for the qualitative phase: how representativeness of the sample will be monitored? For the quantitative phase, please consider that the sample size was calculated using “100% response rate” and this assumption is not supported by results of similar studies. In addition, exclusion criteria described for both women and health professionals during qualitative and quantitative phases are different throughout the manuscript (page 9 [lines 198-205] vs page 19 [lines 404-423]). Please revise and closely edit the section accordingly.

Fourth, authors described that "separate interview guides will be prepared for" health professionals and mothers, but the three additional files cited (line 274) were not available for review. Also, intended validation procedures (line 285) are not clearly described. Please provide more details of procedures, including references and additional files.

Finally, authors outlined as a limitation that the principal investigator are in the upper-level hierarchy of health professionals but they do not clearly specify if PI currently is working at DSHW and/or CSHW. Considering that all in-depth interviews will be conducted by the PI and the possibility of bias, in particular on health professionals’ results, would be helpful to provide few examples of planned mitigating actions to address this limitation.

Reviewer #2: The study hypothesis is certainly of high interest and originality in a context in which the quality of care in obstetrics is not only assessed on the basis of maternal and fetal-neonatal outcomes, but also on the basis of the "perceived" by the patient.

Nonetheless, although the study design is described in detail in the manuscript, I believe it is appropriate to streamline the background inherent the sub-chapters and the methods description because they look too reduntant. In this context I suggest to review the form of the manucript translating some topics from the "Backgorund" to the "Strengths and limitations" session

7. PLOS authors have the option to publish the peer review history of their article (what does this mean?). If published, this will include your full peer review and any attached files.

Reviewer #1: No

Reviewer #2: No

---

## [Author Response · Author response to Decision Letter 0]

22 Nov 2020

Editor’s comments

1. Please ensure that your manuscript meets PLOS ONE's style requirements, including those for file naming. – Thank you. These changes were made in the manuscript in line 1-22.

2. Please include a copy of the interview guide used in the study, in both the original language and English, as Supporting Information, or include a citation if it has been published previously. – Thank you. These changes were made in the manuscript. S2,3,4 Files have been included in line 579-581.

3. Please include additional information regarding the survey or questionnaire used in the study and ensure that you have provided sufficient details that others could replicate the analyses. For instance, if you developed a questionnaire as part of this study and it is not under a copyright more restrictive than CC-BY, please include a copy, in both the original language and English, as Supporting Information.- – Thank you. These changes were made in the manuscript. S5 Files have been included in line 582. However, Part-4: FOBS was not added [we got it from a Sweden author-reference given] and it will be available upon a reasonable request. Two questions in FOBS has mentioned in the manuscript text [437-440]. WP-RMC tool was under copy right from the original authors (Ayoubi, Pazande et al 2020) and could not be included here.

4. Please state whether you validated the questionnaire prior to testing on study participants. Please provide details regarding the validation group within the methods section. – Thank you. These changes were made in the manuscript and detailed validation procedures mentioned in lines 424 for questionnaire-1, 446-469 for FOBS and WP-RMC in 474-498.

5. We note that you have stated that you will provide repository information for your data at acceptance. Should your manuscript be accepted for publication, we will hold it until you provide the relevant accession numbers or DOIs necessary to access your data. If you wish to make changes to your Data Availability statement, please describe these changes in your cover letter and we will update your Data Availability statement to reflect the information you provide. - The data which will be gathered during this this study will be available upon a reasonable request from the corresponding author, not through the repository.

6. Please include a caption for figure 1. . – Thank you. These changes were made in the manuscript in line 175.

7. Please include captions for your Supporting Information files at the end of your manuscript, and update any in-text citations to match accordingly. Please see our Supporting Information guidelines for more information: http://journals.plos.org/plosone/s/supporting-information. . – Thank you. These changes were made in the manuscript in lines 575-584.

8. Additional Editor Comments (if provided):

Please read carefully the feedbacks from the referees and provide a point by point answer. – Thank you. These changes were made in the manuscript and this cover letter.

Please use the appropriate reporting guidelines checklists, such as STROBE and COREQ or SRQR (for qualitative research) Thank you. This was added (SRQR).

Please also specify if the study has been registered in any platform – No, not registered.

Please also specify study timelines – Thank you. These changes were made in the manuscript and a Gantt chart added in lines 532 and 584 and Fig. 2 was also added for Gantt chart.

Reviewers' comments:

Reviewer 1

1. First, the research questions are not clearly stated on last paragraphs of background section, but on lines 102-104 and 113-115; also, research questions are not in line with the following section: Thank you. These changes were made in the manuscript and refer lines 137-141. 

Objectives of the study (line 177). Please revise. Changes were added and in line 164-168. I would suggest the use of reporting guidelines. Checklists, such as STROBE and COREQ or SRQR (for qualitative research). It would help to improve the whole paper comprehension. - Thank you. These changes were made in the manuscript (SRQR was used).

In addition, reference numbering is confusing and not organized in the order that articles were cited in the text (for example, in the first paragraph [lines 78-85] were cited references 8, 11 and 21). I recommend that the authors closely review and edit the paper. -Thank you. These changes were made in the manuscript.

2. Second, on methods would be helpful to provide more information regarding setting and specific hospital units described (example, well-baby clinic). I am assuming that De Soysa Hospital for Women (DSHW) should be also mentioned on lines 467 and 468, please review and edit if necessary. -Thank you. These changes were made in the manuscript. DSHW is also mentioned in lines 186-189.

3. Third, please clarify recruitment procedures. For example, for the qualitative phase: how representativeness of the sample will be monitored? 

-Thank you. Changes were made in the manuscript describing the representativeness. Included in lines 245-252. 

“Participant recruitment will be done while maintaining a diversity to yield a good representativeness during sample selection. Pregnant and postnatal mothers will be recruited starting from uncomplicated cases, then instrumental delivery, elective/emergency caesarean delivery, awaiting a vaginal birth after caesarean (VABC), not willing for a VBAC in this pregnancy, maternal request for a caesarean, and mothers with a history of previous delivery in a different hospital and/or private sector accordingly. For HCP, representativeness will be maintained by selecting general labour ward staff to well-experienced ones, staff from units where labor companion is being practiced, and not being practiced at the moment. 

During the initial introduction and discussion, if investigators feel that the relevant subject is not appropriate, interviewing will be abandoned and another subject will be selected. If the postnatal women are giving insufficient information while staying the hospital, postnatal interviews will be extended into the postnatal well-baby clinics where mothers are expected to give less socially desirable responses as they are not bound to the labour ward anymore. Recruiting and interviewing will be continued until the phenomena under the study are well understood and data saturation is reached”.

4. For the quantitative phase, please consider that the sample size was calculated using “100% response rate” and this assumption is not supported by results of similar studies. 

-Thank you. These changes were made in the manuscript. Considering a 10% non-response rate, a final sample size of 355 was taken as mentioned in line 380.

In addition, exclusion criteria described for both women and health professionals during qualitative and quantitative phases are different throughout the manuscript (page 9 [lines 198-205] vs page 19 [lines 404-423]). Please revise and closely edit the section accordingly. -Thank you. These changes were made in the manuscript. Inclusion and exclusion criteria were corrected [lines 224-232 and 363-364].

4. Fourth, authors described that "separate interview guides will be prepared for" health professionals and mothers, but the three additional files cited (line 274) were not available for review. -Thank you. These changes were made in the manuscript and all the files attached as supporting information (S2,3,4 Files). See lines 579-580.

Also, intended validation procedures (line 285) are not clearly described. Please provide more details of procedures, including references and additional files. -Thank you. These changes were made in the manuscript under the quantitative phase for FOBS [lines 434-460] and WP-RMC scales [lines 446-469]. Refer the manuscript please for these lines as above.

5. Finally, authors outlined as a limitation that the principal investigator are in the upper-level hierarchy of health professionals but they do not clearly specify if PI currently is working at DSHW and/or CSHW. Considering that all in-depth interviews will be conducted by the PI and the possibility of bias, in particular on health professionals’ results, would be helpful to provide few examples of planned mitigating actions to address this limitation. 

-Thank you. These changes were made in the manuscript. Several steps to reduce this effect were added to lines 559-566. 

“PI being in the upper-level of hierarchy of HCP in CSHW has to be acknowledged as a limitation. However, a person with knowledge and experience on obstetric procedures is essential in interviewing as the areas indicate specific medical information and knowledge. Effect of PI will be less at DSHW as he is attached to the CSHW. Data collection by additional medical officers and nurses who are not directly involving with labour care in CSHW and DSHW will also minimize the effect of data collection solely by the PI. Since the study will be to conducted in two hospitals, this potential bias may be minimized with different working and interviewing sites of the investigators”.

Reviewer 2

1. Nonetheless, although the study design is described in detail in the manuscript, I believe it is appropriate to streamline the background inherent the sub-chapters and the methods description because they look too redundant. -Thank you. These changes were made in the manuscript and streamlined accordingly to paragraphs on importance of quality of maternity care, definition and domains of RMC, implications of RMC, Importance of RMC in the Sri Lankan context and justification, aims of the study [lines 79-169 and methods also streamlined].

2. In this context, I suggest to review the form of the manuscript translating some topics from the "Backgorund" to the "Strengths and limitations" session. -Thank you. These changes were made in the manuscript. Please refer the lines 536-554. 

Thank you

Dr.Malitha Patabendige

---

## [Decision Letter · Decision Letter 1]

18 Jan 2021

PONE-D-20-03776R1

Maternal and healthcare providers’ perceptions on respectful maternity care: study protocol for a mixed-methods study.

PLOS ONE

Dear Dr. Patabendige,

Thank you for submitting your manuscript to PLOS ONE. After careful consideration, we feel that it has merit but does not fully meet PLOS ONE’s publication criteria as it currently stands. Therefore, we invite you to submit a revised version of the manuscript that addresses the points raised during the review process.

In this second round of revisions two out of three independent referees recommended rejection. Please read carefully their comments before resubmitting. Please note that resubmission does not guarantee acceptance. 

Kind regards,

Marzia Lazzerini, PhD

Academic Editor

PLOS ONE

Journal Requirements:

 Please check carefully Plos guidelines fore authors 

Reviewers' comments:

Reviewer's Responses to Questions

**Comments to the Author**

1. Does the manuscript provide a valid rationale for the proposed study, with clearly identified and justified research questions?

Reviewer #1: Yes

Reviewer #3: Partly

Reviewer #4: Yes

2. Is the protocol technically sound and planned in a manner that will lead to a meaningful outcome and allow testing the stated hypotheses?

Reviewer #1: No

Reviewer #3: No

Reviewer #4: Yes

3. Is the methodology feasible and described in sufficient detail to allow the work to be replicable?

Reviewer #1: No

Reviewer #3: No

Reviewer #4: Yes

4. Have the authors described where all data underlying the findings will be made available when the study is complete?

Reviewer #1: No

Reviewer #3: No

Reviewer #4: Yes

5. Is the manuscript presented in an intelligible fashion and written in standard English?

Reviewer #1: No

Reviewer #3: No

Reviewer #4: Yes

6. Review Comments to the Author

You may also provide optional suggestions and comments to authors that they might find helpful in planning their study.

Reviewer #1: The current version of the protocol has improved compared with the first one, but, some very important methodological issues were still not appropriately addressed by authors.

Regarding recruitment procedures, ii still not clear how both pregnant or postnatal women would be recruited by clinical characteristics (lines 246 to 250: “uncomplicated cases, instrumental delivery, elective/emergency caesarean delivery, awaiting a vaginal birth after caesarean (VABC), not willing for a VBAC in the current pregnancy, maternal request for a caesarean, and mothers with a history of previous delivery in a different hospital and/or private sector”) without prior access to medical records or privacy violations. For health care professionals, recruitment procedures described on manuscript are vague: it’s not clear the number of professional categories that would be invited to participate and in which proportion those categories will be present in the final sample. In addition, it’s not clear how the specification “units where labor companion is being practiced and not being practiced at the moment” would influence health workers sample size or composition.

Regarding interviews, it is still not clear how interviewer guides both for pregnant/mothers and health care professionals should be used. The documents provided as supplementary files are poor detailed and difficult to be used in a standardized way without additional instructions. Also, the criteria (why and how) “interviewing will be abandoned” (lines 254 and 255) or “extended into postnatal well-baby clinics” (line 257) is very subjective and not clear described on manuscript (add a selection bias for complete interviews?). I also recommend to provide a reference about the “framework approach” that will be used to analyze interviews content (lines 296 and 301) and to provide the checklist file that will be used for quality control of qualitative phase (lines 328-329).

Regarding assessment tools, the questionnaire 1 (intended to collect “maternal opinion on mode of delivery and childbirth”) should be validated in a sample of mothers. According to authors (lines 423-425) the face validity was done “with the participation of obstetricians, public health specialists”. The proposed use of questionnaire 3 (WP-RMC) “for women who were admitted to the labour suite irrespective of mode of delivery” is not in accordance with the original reference n.30 and it’s not clear how further modifications will be performed (lines 486-487). In addition, in line 409 seems that the publication cited on reference number 30 is about Sri Lanka protocol (“changes accepted by the original authors of WP-RMC in Iran (30)”).

The quality of the English throughout the manuscript would be improved by asking a native English-speaking to review the text. The lack of a market copy of the manuscript among the files uploaded by authors have made difficult the review of this version of the manuscript.

Reviewer #3: This manuscript, “Maternal and healthcare providers’ perceptions on respectful maternity care: study

protocol for a mixed-methods study,” describes a planned observational study to use interviews with obstetric patients and providers and interview-led obstetric patient survey assessment of the Fear of Birth Scale to improve understanding of respectful obstetric care in Colombo, Sri Lanka.

First and foremost, I commend the study team for designing a study to help address the very important problem of disrespectful obstetric care, which has been overlooked in much of the world, particularly South Asia. However, I think there is substantial room for improvement in the study design and writing of the paper.

Major concerns:

1) Utility of designing a new obstetric care scale for patients, and based off a scale designed in Sweden. Why not use an existing tool, such as this one that has been validated in India and now used prolifically? Afulani PA, et al. Validation of the person-centered maternity care scale in India. Reproductive Health 2018;15(147). Other major papers in this field are missing, such as https://www.ncbi.nlm.nih.gov/pmc/articles/PMC5764229/ The authors write, “Currently, there are tools have developed to measure the women’s expectations of RMC in global literature.” I think the authors mean no tools, which is untrue. Also the incorrect English of this sentence and throughout the paper really makes it hard to follow, although I am sympathetic to English not being the team’s first language.

2) I have serious concerns about the validity of the interviews. On the provider side, very little detail is given and it’s unclear what is expected to be found. The paper cites a prior study of “poor attitudes” of Sri Lankan obstetricians of allowing labor companions, which is a standard component of respectful maternity care. On the patient side, I strongly disagree that the PI, a male authority figure tied to the women’s care, is the appropriate person to lead all patient interviews and I think it will lead to inaccurate interviews with patients. I strongly think they should be led by a female who is not a healthcare provider – ideally a mother and someone of the same ethnicity.

3) The paper is quite hard to follow – it could be much better organized and I think the reviewer comments could have been more fully addressed.

Specific comments:

Title: Is confusing – I thought when I first read it that this study was about obstetric providers’ perceptions of respectful maternity care. Suggested rephrase to “Perceptions on respectful maternity care in Sri Lanka: Study protocol for a mixed-method study of patients and providers.”

Introduction: Authors conflate “disrespectful maternity care” and “obstetric violence,” which the introduction largely talks about. The Introduction is long and not well focused (4 pages). I would devote more text to the specific Colombo setting. (Example, why is this study needed in addition to obstetric violence and obstetric patient experience studies that have been done in India and other LMICs?)

Qualitative aim: Besides issues mentioned above, I think the study should just interview women postpartum during delivery hospitalization. It does not make sense to interview women hospitalized for delivery who have not yet delivered. HCP is not well defined. I do not agree with only including healthy women with healthy babies – much is to be learned about the experiences of women when complications arise. Unclear why purposive sampling and what is meant by the investigator will “abandon” the interview if he decides the subject is “not appropriate.” This opens up substantial risk of selection bias. Very little detail on the HCP interviews.

Quantitative aim: This aim is overall confusing. I think that the FOBS from the Swedish study will be adapted based on the interviews? In the power calculation, it says the main outcome is “fear of birth” but I have no idea who is being compared. I don’t follow the analysis plan – it’s very vague.

Reviewer #4: PLOS ONE

Maternal and healthcare providers’ perceptions on respectful maternity care: study

protocol for a mixed-methods study.

Manuscript Number: PONE-D-20-03776R1

Article Type: Registered Report Protocol

Full Title: Maternal and healthcare providers’ perceptions on respectful maternity care: study

protocol for a mixed-methods study.

Summary and Impression

This is a research protocol report describing a mixed methods study design for analyzing RMC in Sri Lanka. The report lays out in detail the steps needed to understand and replicate the methods the authors are using. The design of the study answers the study question and contributes new knowledge to the maternal health community.

Major Issues

Major issues were identified during the prior revisions.

Minor Issues

Page 4, Line 96-98. The final two sentences of the paragraph do not flow with the material. Unless there is to be a larger discussion of the global caesarean epidemic, I recommend that both of these sentences can be removed.

Page 6, Line 137-141. This paragraph can be removed and if needed, a summative, single addition added to the list in the prior paragraph’s final sentence.

Page 10, Line 223 (and inclusion criteria overall). Are you unnecessarily limiting the scope of your sample by only including “Birth of a live healthy baby.” If RMC influences outcomes for mothers and newborns, including either newborns with complications or who died during/post-delivery may have substantial value. Please explain your rational for this inclusion criteria in the paper.

Page 10, Line 229. Please provide the rational for excluding HCWs with <1 year of obstetric experience. HCWs with less experience in obstetrics may have an informative viewpoint.

Page 11, Line 256 (and other locations in abstract/text). Please explain how you will identify socially desired responses, if responses are being influenced by the location. Limited responses should be relatively straightforward to detect but responses being influenced by the location more challenging.

Page 12, Line 273. Please make explicit what from the previous items that the “This” is that will minimize potential bias.

Page 13, Line 292. Will all data be independently coded by 2 investigators? Later in the manuscript it seems like it will however, this should be clarified here and how any disagreements will be resolved.

Page 13, Line 308-314. The citation is enough, and you can remove the list.

Page 14, Linen 319-320. The first sentence is repetitive with subsection title and not needed.

Page 14, Line 328. Add brief statement on how the interview guides and note-taker forms will be pretested.

Page 14, Line 335-336. #6 does not have detail and therefore does not add value. Suggest removing.

Page 15, Line 356-360. Inclusion criteria for the quantitative phase. Are women not able to read sufficiently, excluded from the quantitative portion as two parts are self administered. If so, please add to Limitations. If not, please explain how this will be handled.

Page 16, Line 377. Typo please correct (Z&)

Page 16, Line 380. Please provide a reference if available to the anticipated 10% non-response rate. This seems low for this vulnerable population.

Page 21, Line 486-. Translation is described repeatedly in the manuscript. This may be better consolidated into a single area in the methods section for brevity.

Page 23, Line 537. Please succinctly state what the research gap is specifically.

Page 24, Line 559. The line “However, proving proper information might reduce this” is unclear.

Page 23-24. Limitations. Is having a male interviewer a potential limitation? Should this be explicitly stated?

Overall, please check carefully for typos and grammar.

7. PLOS authors have the option to publish the peer review history of their article (what does this mean?). If published, this will include your full peer review and any attached files.

Reviewer #1: No

Reviewer #3: No

Reviewer #4: **Yes: **Matthew C. Strehlow

---

## [Author Response · Author response to Decision Letter 1]

6 Mar 2021

PLOS ONE

Perceptions on respectful maternity care in Sri Lanka: Study protocol for a mixed-methods study of patients and providers.

Manuscript Number: PONE-D-20-03776R1

Article Type: Registered Report Protocol

My sincere gratitude goes to the editors and reviewers for their generous effort in improving this manuscript. All the comments raised by the editors and reviewers have been addressed point by point.

Thank you

Dr.Malitha Patabendige

Reviewer #1: 

The current version of the protocol has improved compared with the first one, but, some very important methodological issues were still not appropriately addressed by authors.

Regarding recruitment procedures, ii still not clear how both pregnant or postnatal women would be recruited by clinical characteristics (lines 246 to 250: “uncomplicated cases, instrumental delivery, elective/emergency caesarean delivery, awaiting a vaginal birth after caesarean (VABC), not willing for a VBAC in the current pregnancy, maternal request for a caesarean, and mothers with a history of previous delivery in a different hospital and/or private sector”) without prior access to medical records or privacy violations. 

– Thank you, these controversial issues have been addressed.

For health care professionals, recruitment procedures described on manuscript are vague: it’s not clear the number of professional categories that would be invited to participate and in which proportion those categories will be present in the final sample. In addition, it’s not clear how the specification “units where labor companion is being practiced and not being practiced at the moment” would influence health workers sample size or composition. 

– Thank you, these controversial issues have been addressed. Details have been included about the HCP and how to deal with units without labour companionship.

Regarding interviews, it is still not clear how interviewer guides both for pregnant/mothers and health care professionals should be used. The documents provided as supplementary files are poor detailed and difficult to be used in a standardized way without additional instructions. 

– Thank you, these controversial issues have been addressed. I personally contacted Prof.Afulani and went through her interview guides and learnt a lot to improve interviewer guides.

Also, the criteria (why and how) “interviewing will be abandoned” (lines 254 and 255) or “extended into postnatal well-baby clinics” (line 257) is very subjective and not clear described on manuscript (add a selection bias for complete interviews?). 

– Thank you, these controversial issues have been addressed and removed.

I also recommend to provide a reference about the “framework approach” that will be used to analyze interviews content (lines 296 and 301) and to provide the checklist file that will be used for quality control of qualitative phase (lines 328-329).

– Thank you, these controversial issues have been addressed. References provided.

Regarding assessment tools, the questionnaire 1 (intended to collect “maternal opinion on mode of delivery and childbirth”) should be validated in a sample of mothers. According to authors (lines 423-425) the face validity was done “with the participation of obstetricians, public health specialists”. The proposed use of questionnaire 3 (WP-RMC) “for women who were admitted to the labour suite irrespective of mode of delivery” is not in accordance with the original reference n.30 and it’s not clear how further modifications will be performed (lines 486-487). In addition, in line 409 seems that the publication cited on reference number 30 is about Sri Lanka protocol (“changes accepted by the original authors of WP-RMC in Iran (30)”).

– Thank you, these controversial issues have been addressed. Pre-testing of questionnaire 01 among a sample of 25 postnatal women has been done.

The quality of the English throughout the manuscript would be improved by asking a native English-speaking to review the text. The lack of a market copy of the manuscript among the files uploaded by authors have made difficult the review of this version of the manuscript.

– Thank you, copy with tracked-changes has been attached for your reference.

Reviewer #3: 

This manuscript, “Maternal and healthcare providers’ perceptions on respectful maternity care: study protocol for a mixed-methods study,” describes a planned observational study to use interviews with obstetric patients and providers and interview-led obstetric patient survey assessment of the Fear of Birth Scale to improve understanding of respectful obstetric care in Colombo, Sri Lanka.

First and foremost, I commend the study team for designing a study to help address the very important problem of disrespectful obstetric care, which has been overlooked in much of the world, particularly South Asia. However, I think there is substantial room for improvement in the study design and writing of the paper.

Major concerns:

1) Utility of designing a new obstetric care scale for patients, and based off a scale designed in Sweden. Why not use an existing tool, such as this one that has been validated in India and now used prolifically? Afulani PA, et al. Validation of the person-centered maternity care scale in India. Reproductive Health 2018;15(147). Other major papers in this field are missing, such as https://www.ncbi.nlm.nih.gov/pmc/articles/PMC5764229/ The authors write, “Currently, there are tools have developed to measure the women’s expectations of RMC in global literature.” I think the authors mean no tools, which is untrue. Also the incorrect English of this sentence and throughout the paper really makes it hard to follow, although I am sympathetic to English not being the team’s first language.

– Thank you, these important issues have been addressed. References have been added. Justification has provided why we prefer Iranian tool- WP-RMC. We have also acknowledged the value of important work by Afulani et al on RMC.

2) I have serious concerns about the validity of the interviews. On the provider side, very little detail is given and it’s unclear what is expected to be found. The paper cites a prior study of “poor attitudes” of Sri Lankan obstetricians of allowing labor companions, which is a standard component of respectful maternity care. On the patient side, I strongly disagree that the PI, a male authority figure tied to the women’s care, is the appropriate person to lead all patient interviews and I think it will lead to inaccurate interviews with patients. I strongly think they should be led by a female who is not a healthcare provider – ideally a mother and someone of the same ethnicity.

– Thank you, we agree with your concerns and a female group of data collectors was included. Details added for the providers’ side.

3) The paper is quite hard to follow – it could be much better organized and I think the reviewer comments could have been more fully addressed.

- Thank you, the entire manuscript was re-written and completely changed.

Specific comments:

Title: Is confusing – I thought when I first read it that this study was about obstetric providers’ perceptions of respectful maternity care. Suggested rephrase to “Perceptions on respectful maternity care in Sri Lanka: Study protocol for a mixed-method study of patients and providers.”

- Thank you, we have changed the title as suggested.

Introduction: Authors conflate “disrespectful maternity care” and “obstetric violence,” which the introduction largely talks about. The Introduction is long and not well focused (4 pages). I would devote more text to the specific Colombo setting. (Example, why is this study needed in addition to obstetric violence and obstetric patient experience studies that have been done in India and other LMICs?)

- Sri Lanka is a unique setting among other LMIC, so then Colombo setting has been justified.

Qualitative aim: Besides issues mentioned above, I think the study should just interview women postpartum during delivery hospitalization. It does not make sense to interview women hospitalized for delivery who have not yet delivered. HCP is not well defined. I do not agree with only including healthy women with healthy babies – much is to be learned about the experiences of women when complications arise. 

- Thank you, we have included only the vaginally delivered postnatal women. Women who had complications also included. HCP and the aims were defined.

Unclear why purposive sampling and what is meant by the investigator will “abandon” the interview if he decides the subject is “not appropriate.” This opens up substantial risk of selection bias. Very little detail on the HCP interviews.

- Thank you, these have been addressed completely. Controversial points [investigator will “abandon” the interview if he decides the subject is “not appropriate] have been removed. 

Quantitative aim: This aim is overall confusing. I think that the FOBS from the Swedish study will be adapted based on the interviews? In the power calculation, it says the main outcome is “fear of birth” but I have no idea who is being compared. I don’t follow the analysis plan – it’s very vague. 

- Thank you, we have removed the FOBS from the protocol and focused only on the RMC. Accordingly, analysis plan was made clear.

Reviewer #4: PLOS ONE

Maternal and healthcare providers’ perceptions on respectful maternity care: study protocol for a mixed-methods study.

Summary and Impression

This is a research protocol report describing a mixed methods study design for analyzing RMC in Sri Lanka. The report lays out in detail the steps needed to understand and replicate the methods the authors are using. The design of the study answers the study question and contributes new knowledge to the maternal health community.

Major Issues

Major issues were identified during the prior revisions.

- Thank you.

Minor Issues

Page 4, Line 96-98. The final two sentences of the paragraph do not flow with the material. Unless there is to be a larger discussion of the global caesarean epidemic, I recommend that both of these sentences can be removed.

- Thank you, we have addressed these issues and the manuscript was updated accordingly.

Page 6, Line 137-141. This paragraph can be removed and if needed, a summative, single addition added to the list in the prior paragraph’s final sentence.

- Thank you, we have addressed these issues and the manuscript was updated accordingly.

Page 10, Line 223 (and inclusion criteria overall). Are you unnecessarily limiting the scope of your sample by only including “Birth of a live healthy baby.” If RMC influences outcomes for mothers and newborns, including either newborns with complications or who died during/post-delivery may have substantial value. Please explain your rational for this inclusion criteria in the paper.

- Thank you, we have addressed these issues and the manuscript was updated accordingly. Women with complications also added.

Page 10, Line 229. Please provide the rational for excluding HCWs with <1 year of obstetric experience. HCWs with less experience in obstetrics may have an informative viewpoint. 

- Thank you, we have addressed these issues and the manuscript was updated accordingly. They have also included.

Page 11, Line 256 (and other locations in abstract/text). Please explain how you will identify socially desired responses, if responses are being influenced by the location. Limited responses should be relatively straightforward to detect but responses being influenced by the location more challenging. 

- Thank you, we have addressed these issues and the manuscript was updated accordingly. This could be a limitation.

Page 12, Line 273. Please make explicit what from the previous items that the “This” is that will minimize potential bias.

- Thank you, we have addressed these issues and the manuscript was updated accordingly. Controversial points removed.

Page 13, Line 292. Will all data be independently coded by 2 investigators? Later in the manuscript it seems like it will however, this should be clarified here and how any disagreements will be resolved.

- Thank you, we have addressed these issues and the manuscript was updated accordingly.

Page 13, Line 308-314. The citation is enough, and you can remove the list.

Page 14, Linen 319-320. The first sentence is repetitive with subsection title and not needed.

Page 14, Line 328. Add brief statement on how the interview guides and note-taker forms will be pretested. 

- Thank you, we have addressed these issues and the manuscript was updated accordingly.

Page 14, Line 335-336. #6 does not have detail and therefore does not add value. Suggest removing. 

Page 15, Line 356-360. Inclusion criteria for the quantitative phase. Are women not able to read sufficiently, excluded from the quantitative portion as two parts are self-administered. If so, please add to Limitations. If not, please explain how this will be handled.

- Thank you, we have addressed these issues and the manuscript was updated accordingly.

Page 16, Line 377. Typo please correct (Z&)

Page 16, Line 380. Please provide a reference if available to the anticipated 10% non-response rate. This seems low for this vulnerable population. 

Page 21, Line 486-. Translation is described repeatedly in the manuscript. This may be better consolidated into a single area in the methods section for brevity.

- Thank you, we have addressed these issues and the manuscript was updated accordingly.

Page 23, Line 537. Please succinctly state what the research gap is specifically.

Page 24, Line 559. The line “However, proving proper information might reduce this” is unclear.

Page 23-24. Limitations. Is having a male interviewer a potential limitation? Should this be explicitly stated?. Overall, please check carefully for typos and grammar. 

- Thank you, we have addressed these issues and the manuscript was updated accordingly.

---

## [Decision Letter · Decision Letter 2]

19 Apr 2021

Perceptions on respectful maternity care in Sri Lanka: Study protocol for a mixed-methods study of patients and providers.

PONE-D-20-03776R2

Dear Dr. Patabendige,

We’re pleased to inform you that your manuscript has been judged scientifically suitable for publication and will be formally accepted for publication once it meets all outstanding technical requirements.

Kind regards,

Tanya Doherty, PhD

Academic Editor

PLOS ONE

Additional Editor Comments (optional):

Reviewers' comments:

Reviewer's Responses to Questions

**Comments to the Author**

1. Does the manuscript provide a valid rationale for the proposed study, with clearly identified and justified research questions?

Reviewer #3: Yes

2. Is the protocol technically sound and planned in a manner that will lead to a meaningful outcome and allow testing the stated hypotheses?

Reviewer #3: Yes

3. Is the methodology feasible and described in sufficient detail to allow the work to be replicable?

Reviewer #3: Yes

4. Have the authors described where all data underlying the findings will be made available when the study is complete?

Reviewer #3: Yes

5. Is the manuscript presented in an intelligible fashion and written in standard English?

Reviewer #3: Yes

6. Review Comments to the Author

You may also provide optional suggestions and comments to authors that they might find helpful in planning their study.

Reviewer #3: I am satisfied with the changes made in response to the reviewer comments. I have no further comments to add at this point.

7. PLOS authors have the option to publish the peer review history of their article (what does this mean?). If published, this will include your full peer review and any attached files.

Reviewer #3: No

---

## [Editor Report · Acceptance letter]

23 Apr 2021

PONE-D-20-03776R2 

Perceptions on respectful maternity care in Sri Lanka: Study protocol for a mixed-methods study of patients and providers. 

Dear Dr. Patabendige:

I'm pleased to inform you that your manuscript has been deemed suitable for publication in PLOS ONE. Congratulations! Your manuscript is now with our production department. 

Kind regards, 

on behalf of

Professor Tanya Doherty 

Academic Editor

PLOS ONE